# Structural and Functional Neural Correlates in Individuals with Excessive Smartphone Use: A Systematic Review and Meta-Analysis

**DOI:** 10.3390/ijerph192316277

**Published:** 2022-12-05

**Authors:** Hsiu-Man Lin, Yu-Tzu Chang, Meng-Hsiang Chen, Shu-Tsen Liu, Bo-Shen Chen, Lin Li, Chiao-Yu Lee, Yu-Ru Sue, Tsai-Mei Sung, Cheuk-Kwan Sun, Pin-Yang Yeh

**Affiliations:** 1Division of Child and Adolescent Psychiatry & Division of Developmental and Behavioral Pediatrics, China Medical University Children’s Hospital, Taichung 404327, Taiwan; 2School of Post Baccalaureate Chinese Medicine, China Medical University, Taichung 406040, Taiwan; 3Division of Pediatric Neurology, China Medical University Children’s Hospital, Taichung 404327, Taiwan; 4Department of Diagnostic Radiology, Kaohsiung Chang Gung Memorial Hospital, Kaohsiung 83301, Taiwan; 5College of Medicine, Chang Gung University, Kaohsiung 83300, Taiwan; 6Department of Psychology, College of Medical and Health Science, Asia University, Taichung 41354, Taiwan; 7Department of Emergency Medicine, E-Da Hospital, I-Shou University, Kaohsiung 824005, Taiwan; 8School of Medicine for International Students, I-Shou University, Kaohsiung 82445, Taiwan; 9Clinical Psychology Center, Asia University Hospital, Taichung 41354, Taiwan

**Keywords:** smartphone addiction, excessive smartphone use, problematic smartphone use, brain volume, functional connectivity, magnetic resonance imaging

## Abstract

Background: Despite known association of internet addiction with a reduced brain volume and abnormal connectivity, the impact of excessive smartphone use remains unclear. Methods: PubMed, Embase, ClinicalTrial.gov, and Web of Science databases were systematically searched from inception to July 2022 using appropriate keywords for observational studies comparing differences in brain volumes and activations between excessive smartphone users and individuals with regular use by magnetic resonance imaging. Results: Of the 11 eligible studies retrieved from 6993 articles initially screened, seven and six evaluated brain volumes and activations, respectively. The former enrolled 421 participants (165 excessive smartphone users vs. 256 controls), while the latter recruited 276 subjects with 139 excessive smartphone users. The results demonstrated a smaller brain volume in excessive smartphone users compared to the controls (*g* = −0.55, *p* < 0.001), especially in subcortical regions (*p* < 0.001). Besides, the impact was more pronounced in adolescents than in adults (*p* < 0.001). Regression analysis revealed a significant positive association between impulsivity and volume reduction. Regarding altered activations, the convergences of foci in the declive of the posterior lobe of cerebellum, the lingual gyrus, and the middle frontal gyrus were noted. Conclusions: Our findings demonstrated a potential association of excessive smartphone use with a reduced brain volume and altered activations.

## 1. Introduction

With the prevalence of smartphones worldwide, their excessive use has already become a social issue. In contrast to other forms of addiction such as gaming or gambling addiction that has been categorized as a distinct disease entity according to the International Classification of Disease (ICD) [1,2], excessive smartphone use is a more general behavioral addiction that has not been officially classified as a disorder [3]. Compared with drug dependence that affects structural and functional neural correlates through chemical pathways, changes associated with behavioral addiction are more likely through operant learning that involves rewards and punishments for behavioral impacts [4,5,6]. Excessive smartphone use has gradually replaced internet addiction as the most common behavioral addiction because of the need for communication and the convenience of its use [7]. Previous studies mainly addressed issues surrounding internet addiction [8] without focusing on the use of smartphones, which have become an indispensable communicating device in over 83% of the global population [8]. Although a previous meta-analysis has reported a reduction in gray matter volume (GMV) as well as significant activations in the medial/superior frontal gyri, the left anterior cingulate cortex/cingulate gyrus, and the left middle frontal/precentral gyri in individuals with internet addiction [9], the structural and functional impacts of excessive smartphone use on the central nervous system remains unclear. Therefore, the current meta-analysis aimed at investigating the difference in structural volume and functional connectivity between individuals with excessive smartphone use and their comparators with regular use.

Magnetic resonance imaging (MRI) is the most popular assessment tool for macroscopic neuroanatomical assessment of the brain (e.g., brain volume measurement) because of its excellent image resolution and quality of between-tissue contrast [10]. Consistently, MRI has been used to examine anatomic differences (e.g., gray matter volume) among individuals with behavioral addiction [9,11]. MRI systems of different magnetic strengths have their distinct merits in MRI acquisition. While 3-Tesla systems per se offer an improved resolution of between-tissue contrast, MRI scan using 1.5-Tesla systems has the advantage of providing sufficient data on quantifying relatively small brain structures [10]. Besides, several types of MR sequences are available for structural neuroimaging, such as T1-weighted and T2-weighted imaging. In contrast to T2-weighted imaging, the T1-weighted approach that provides the greatest clarity in distinguishing among gray matter, white matter, and cerebrospinal fluid (CSF) is frequently used for quantitative MRI studies of brain morphology (i.e., individual brain structures) [10]. Regarding the computation of structural volume, voxel-wise statistical analysis (e.g., voxel-based morphometry) [12] is commonly used. It is a fully automated MR image analysis technique that permits voxel-wise statistical comparison of the local concentration of GMV between two groups of subjects (e.g., patients vs. healthy controls). Hence, the current study focused on the use of voxel-based structural brain imaging for volumetric measurement (mm^3^) in assessing the difference in brain volume between individuals with excessive smartphone use and healthy controls. Furthermore, the associations of moderators (i.e., stages of life and brain areas) and mediators (i.e., demographic variables) with brain volume were examined and analyzed.

In addition to computing the structural neural correlate, we also combined coordinate-based meta-analysis to identify regions of consistent activation across functional MRI (fMRI) studies, followed by the determination of the convergence of foci reported from different studies (or experiments) using activation likelihood estimation (ALE) [13]. The current meta-analysis attempted to integrate brain structures with functional connectivity to provide neurobiological information about the differences between individuals with excessive smartphone use and healthy comparators.

## 2. Methods

### 2.1. Study Eligibility

The excellent resolution of MRI in differentiation among different structures makes it an ideal tool for discerning subtle anatomical differences in cortical and subcortical structures [10]. Regarding the evaluation of functional connectivity, fMRI is often used because of its ability to measure metabolic activity within anatomic structures. The eligibility criteria for study inclusion were: (i) Original, cross-sectional comparative studies that used MRI in individuals with excessive smartphone use versus their counterparts with regular use; (ii) Recruitment of individuals with excessive smartphone use confirmed through a psychiatric interview or a validated questionnaire; (iii) Studies that used a whole-brain or region of interest analysis; (v) those that reported the differences in structural volumes or peak coordinates between participants with excessive smartphone use and control groups. On the other hand, we excluded studies that focused on “non-smart” mobile phones as well as those involved participants with brain injury (e.g., traumatic or stroke), severe mental disorders, neurological or physical diseases, and those examining the treatment effects of drugs or non-drugs.

### 2.2. Electronic Searches

Following the PRISMA guidelines [14] (Appendix A) and the MOOSE statement [15], we systematically searched for articles through the PubMed, Embase, ClinicalTrials.gov and Web of Science databases from inception to July 2022. For completeness of our literature search, we used different string terms, namely mobile phone addiction (or internet addiction or smartphone addiction or problematic phone use or overuse phone or nomophobia or excess phone use or phone addictive behaviors or pathological phone use or phone abuse) AND imaging (or magnetic resonance imaging or MRI or VBM or voxel-based morphometry) (Appendix A). No restriction was placed on language, date of publication, and country of origin. The systematic review protocol was registered on the International Prospective Register of Systematic Reviews (PROSPERO) website (ID 359367).

### 2.3. Data Extraction

Five independent authors (Chen B.-S., Li L., Lee C.-Y., Sue Y.-R. and Sung T.-M.) completed title and abstract screening. Two other authors (Yeh P.-Y. and Liu S.-T.) independently screened the full text of the retrieved literature with disagreements resolved through discussion until consensus was reached. On encountering articles in which the necessary data were unavailable, we attempted to retrieve the information through electronically contacting the corresponding authors. For different studies using the same data, the article with more recent information or a larger sample size was selected for the current investigation.

### 2.4. Effect Size Analysis

For structural neuroimaging data synthesis, effect size (ES) was used as the measure for the primary outcome (i.e., brain volume) across the included studies. The current meta-analysis used Hedges’ *g* as the ES which was the difference in mean structural volumes between individuals with excessive smartphone use and their comparators with regular use divided by the pooled standard deviations of the two groups. To compute the ESs, we used the software “Comprehensive Meta-Analysis version for Windows (CMA, version 3.0)”. A negative ES represented a smaller brain volume in individuals with excessive smartphone use compared to their counterparts with regular use. If more than one dataset (e.g., different brain regions) were available in the eligible studies, a single ES denoting the mean volume was acquired through standardization and averaging of the results. In respect of the significance of findings, ESs of 0.8, 0.5, and 0.2 were deemed large, moderate, and small, respectively [16].

We used subgroup analyses to evaluate the effects of moderators on brain volume. Because the relatively small sample sizes commonly encountered in neuroimaging studies would decrease statistical power, subgroup analysis in the current meta-analysis was based on a random-effects model [17] on the assumption of an average distribution of ESs across the included studies [18] to minimize sample size bias. In contrast to the weights assessed with the fixed-effects model, those in the random-effects model are more similar to each other [19]. We also used *Q* statistics and the corresponding *p* values to detect the heterogeneity of ESs. As for the examination of the association between the mediators (e.g., age and prevalence of females) and brain volume, meta-regression with a mixed-effects model was performed.

We used different methods to evaluate publication bias according to the number of eligible studies. For outcomes reported in fewer than ten datasets, the corresponding funnel plots were inspected [20]. Egger’s regression test was used for outcomes described in ten or more datasets [21]. Using the Duval and Tweedie’s trim and fill method, potentially missing studies were imputed for evidence of funnel plot asymmetry [22]. The influence of individual studies on the overall outcome was assessed with leave-one-out sensitivity analysis [23].

### 2.5. ALE Analysis

ALE analysis was conducted with GingerALE v3.0.2 in the BrainMap environment [13,24,25]. There are two major statistical approaches to image analysis acquired with MRI, namely, cluster-based approach (CBA) and voxel-based approach (VBA). While CBA can detect a larger proportion of truly active brain areas (i.e., higher sensitivity to activation detection with large spatial extent) especially studies with moderate effect sizes and smaller sample sizes [26,27], VBA is suggested to avoid pitfalls [27] despite being too conservative to decrease power [28]. Therefore, CBA is recommended for ALE studies [28]. In addition, we used CBA to identify relative dominance because of the small number of eligible studies and their small sample sizes. Coordinate data were entered into Talairach space. Coordinates reported in MNI space were converted to Talairach space based on the icbm2tal transform [29], followed by GingerALE calculations. Modeled Activation (MA) maps were created for each foci group by modeling with Gaussians distribution. ALE map was constructed by combining a union of all of the MA maps with a table of *p*-value for ALE score. The ALE image and *p*-value table were used to create a 3D *p*-value image. Using the GingerALE software, the threshold of *p*-value image was then set utilizing the family-wise error (FWE) method with *p* < 0.05 being defined as statistically significant. We performed two separate ALE analyses in this study for two different conditions, namely, increased and decreased activations among individuals with excessive smartphone use compared to controls. Significant clusters were overlaid onto an anatomical Talairach template, colin_tlrc_2x2x2.nii (http://www.brainmap.org/ale, accessed on 16 May 2022), using the Mango software (version 4.1, Research Imaging Institute, University of Texas Health Science Center, Houston, TX, USA; http://www.ric.uthscsa.edu/mango, accessed on 16 May 2022).

## 3. Results

### 3.1. Study Characteristics and Participants

Figure 1 depicts the process of identification of eligible studies in the current meta-analysis. Of the 6993 full-text articles deemed eligible according to their abstracts and titles, 6982 were excluded because of failure to meet the inclusion criteria. Further, two studies were not qualified for this study [30,31] since they shared the same participants with the study by Horvath et al. [32]. Finally, 11 articles were included in this meta-analysis (Table 1). The definitions of excessive (problematic) smartphone users across the included studies are summarized in Appendix A. Of the 11 articles, nine used questionnaires for excessive smartphone use including the smartphone addiction proneness scale (SAPS) (*n* = 5), smartphone addiction scale (SAS) (*n* = 3), and Mobile phone addiction index (MPAI) (*n* = 1) for evaluating the severity of excessive smartphone use as well as grouping participants into individuals with excessive smartphone use and comparators with regular use (Table 1). However, of the other two articles, one used the Facebook version of the compulsive internet use scale (CIUS) and the other adopted the Young’s internet addiction test (Table 1).

Of the 11 included articles, seven compared the brain volumes between individuals with excessive smartphone use (*n* = 165) and their comparators with regular use (*n* = 256) (Table 1). The mean age of the participants was 19.61 years (range, 15.26 to 29.80 years) with a female prevalence of 52.53% (range, 25.56.7 to 68.18%) (Table 1). With reference to brain activations, six studies investigated the functional connectivity of individuals with excessive smartphone use (*n* = 139, mean age = 24.18 years) and their counterparts with regular use (*n* = 137, mean age = 24.47 years) based on ALE analyses of their fMRI findings (Table 1). Because excessive smartphone use was found to be associated with activations of certain brain regions but decreased activities in others, data were collected from four studies that provided information about the regions with enhanced activities among excessive users compared to regular users (176 participants, 96 activation foci) [34,35,37,39] and four studies reporting the regions with suppressed activities (196 participants, 25 activation foci) [32,34,37,40].

Some studies used self-reported evaluation tools to investigate the severity of impulsivity (*n* = 8), depression (*n* = 6), and anxiety (*n* = 5). Focusing on the assessment of impulsivity, four studies [32,35,38,42] used the Barratt impulsiveness scale (BIS) [43] with 30 items, each of which was given a Likert score of 0 to 3 to indicate a progressive increase in severity. Regarding the evaluation of depression, three studies [32,33,38] adopted the Beck depression inventory (BDI) [44], which assesses the severity of depression based on the participant’s responses to 21 items, each of which is scored on a four-point Likert scale of 0, 1, 2, and 3 representing no, mild, moderate, and severe depression, respectively. In contrast, the five studies with self-reported information about anxiety used different scales for assessment that precluded the subsequent conduction of regression analysis. 

The 11 eligible clinical trials were conducted mostly in Asian countries including Korea (*n* = 5), China (*n* = 3), and Malaysia (*n* = 2), while one study was performed in Germany. The sample size of the included studies ranged from 30 to 88.

### 3.2. Quantitative Brain Volume Data Synthesis

The results from seven studies demonstrated a significant anatomical difference between individuals with excessive smartphone use and their comparators with regular use (Hedges’ *g* = −0.55, 95% CI = −0.80 to −0.31, *p* < 0.001) (Figure 2), indicating a decreased brain volume in individuals with excessive smartphone use. The ES was unchanged on leave-one-out sensitivity analysis (*p* < 0.001), suggesting a non-significant impact of the results of any single study on the main outcome. The risks of publication bias reflected by funnel plot asymmetry for brain volume is shown in Figure 3. Because the ‘trim and fill’ method revealed zero potentially missing studies on the left side of the plot, the adjusted ES of −0.55 (−0.80 to −0.31) remained unchanged with the random-effects model.

Subgroup analysis focusing on age groups among individuals with excessive smartphone use (seven studies, 421 participants) revealed a significantly decreased brain volume in adolescents compared to the adults (*p* < 0.001), suggesting a more pronounced structural impact of excessive smartphone use among adolescents (Table 2). Subgroup analysis for cortical and subcortical regions in individuals with excessive smartphone use showed a significant reduction in size of the subcortical structures compared to that of the cerebral cortex (*p* < 0.001) (Table 2), implicating an association between excessive smartphone use and a reduced volume of subcortical structures. The results of meta-regression analysis demonstrated no correlation between most of the investigated mediators and brain volume. However, a significant correlation was noted between the BIS score and brain volume in individuals with excessive smartphone use (*p* = 0.03; Table 3), suggesting a reduced volume in those with a higher impulsivity.

### 3.3. The Difference in Brain Activations between Individuals with Excessive Smartphone Use and Their Counterparts with Regular Use

ALE analysis of the four studies [34,35,37,39] with available information about the anatomical locations with an enhancement of brain activities among individuals with excessive smartphone use showed consistent areas of activations, namely the declive of posterior lobe of the cerebellum, the lingual gyrus (BA 18), and the middle frontal gyrus (BA 46) (Table 4 and Figure 4). On the other hand, analysis of the four studies [32,34,37,40] providing data on the anatomical regions with suppressed activities in excessive cellphone users failed to pinpoint commonly affected areas.

## 4. Discussion

To our knowledge, the current study is the first meta-analysis to investigate the effect of excessive smartphone use on the structural and functional neural correlates. Our investigation revealed several findings that would be of clinical interest. First, individuals with excessive smartphone use showed a reduced brain volume, especially among the subcortical structures. Second, the impact on brain volume was more pronounced in adolescents than in adults. Third, our finding of a positive association between impulsivity and a reduced volume suggested a structural impact of impulsiveness on the brain. Fourth, our identification of significant spatial convergence in the declive of posterior lobe, the lingual gyrus (BA 18), and the middle frontal gyrus (BA 46) in individuals with excessive smartphone use indicated an anterior-posterior distribution of altered connectivity.

Focusing on the effects of excessive smartphone use on structural and functional neural correlates, the current study yielded several interesting results. Our finding of a relationship between excessive smartphone use and a decreased brain volume, especially subcortical structures, was consistent with that of a previous meta-analysis that demonstrated a correlation between internet addiction and a reduced GMV [9]. One of the possible mechanisms underlying the observation may be the emotional impacts associated with excessive smartphone use. Indeed, a body of evidence from systematic reviews has revealed a significant association between problematic smartphone use and mental health with anxiety and depression being the most common comorbid mental diseases [7,45]. In continuation with this finding, previous meta-analyses further showed a reduced GMV in the bilateral anterior cingulate cortex (ACC) among patients with major depressive disorder [46] as well as decreased volumes of the right ventral ACC and the left inferior frontal gyrus in individuals with anxiety disorders [47]. Besides anxiety and depression, other mental disorders have also been shown to be associated with a reduction in volume of the dorsal ACC and the bilateral insula [48].

Our ALE results revealed an association between excessive smartphone use and activation of several brain areas, namely the right posterior lobe of cerebellum, the right lingual gyrus (BA18), and the left middle frontal gyrus (BA46) (Figure 4 and Table 4). Regarding the relationship between the cerebellum and behavioral addiction, a previous observational study has reported a disruption of executive-cerebellar networks but increased occipital-putamen connectivity in internet addicts, probably resulting from addiction-sensitive cognitive control processes and bottom-up sensory plasticity [49]. With respect to the possible role of the lingual gyrus (BA18) in excessive smartphone use, a previous investigation has demonstrated that not only does the area receive information from the primary visual cortex but it also affects the cortical endophenotypes for anxiety and depression [50] which are common emotional comorbidities related to problematic smartphone use [7,45]. Consistent with the function of automatic emotion regulation of the cerebellum [49] as well as the role of the lingual gyrus (BA18) in modulating both voluntary and automatic emotions [50], a previous clinical trial has shown that the middle frontal gyrus (BA46) is also responsible for voluntary and automatic emotion control [51].

Another interesting finding of the present study was the variation in structural impacts of excessive smartphone use in different age groups on the brain. It has been reported that adolescents are more prone to excessive smartphone use compared to adults [52,53]. Previous studies on adolescents based on voxel-based morphometry demonstrated a lower GMV in multiple regions among those with internet addiction (especially male) [54,55], including in the bilateral dorsolateral prefrontal cortex, the supplementary motor area, the orbitofrontal cortex, the cerebellum, and the left rostral ACC. In addition to supporting the previous finding that addictive behavior is associated with a smaller GMV [54,55], our results further demonstrated a similar impact of excessive smartphone use as well as a more obvious effect in adolescents than in adults (Table 2). A change in GMV has been found to have a functional impact. In contrast to a reduction in GMV and cortical thickness from childhood to young adulthood, gray matter density (GMD) increases during this period [56] from GMV transformation to increase neuroreceptor and neurotransporter availability [57]. As there is a positive correlation between GMD and neurocognition [58], a smaller GMV in adolescents with excessive smartphone use may influence GMD and thereby weaken neurocognitive performance. Furthermore, our meta-regression analysis revealed a significant positive correlation between the BIS score (i.e., impulsivity levels) and a decreased brain volume (Table 3). Our finding was consistent was that of previous studies that reported a smaller brain volume in individuals diagnosed with depression, anxiety, or attention-deficit/hyperactivity disorder (ADHD) [7,45], taking into account the known positive associations of these disorders with impulsivity and excessive smartphone use [59]. Taken together, emotion dysregulation may play an important role in neurological development among adolescent problematic smartphone users.

One of the therapeutic implications of the present investigation is a restoration of brain volume. Given the low probability of a reduction in smartphone use, other measures may be taken. Neurofeedback (NF), which is a well-accepted technique for regulating brain activities, has been found to alter brain microstructures [60]. Nevertheless, a significant limitation of EEG-based NF is its confinement to the detection and modification of cortical instead of subcortical activities. In contrast, real-time fMRI-based NF, which can discern whole-brain activities, has been used to restore the size of hippocampus in patients with post-traumatic stress disorder through amplifying the NF signals from the left amygdala [61]. In addition, compared to fMRI-based NF, low-resolution electromagnetic tomography (LORETA) NF is another more economical approach based on the theory of brain volume conduction [62]. LORETA NF could be regarded as a form of neuropsychotherapy [63] to more accurately restore the reduced volume and downregulate the activity of neural mediators related to excessive smartphone use (i.e., BA 18 and BA46; Table 4).

The current meta-analysis had its limitations. First, because our findings of structural and functional neural correlates were based on data from a limited number of available studies (i.e., seven and six, respectively), further investigations are needed to support our findings. Second, heterogeneity in task-evoked stimuli (i.e., social media, entertainment, game playing) being used across the eligible fMRI studies may introduce bias to the localization of the brain areas being affected. Nevertheless, we identified consistent brain regions showing activations despite failure to specify the regions with decreased activities. Third, the current meta-analysis revealed a publication bias; while there were five studies on the left side of the funnel plot in support of a reduction in brain volume among individuals with excessive smartphone use (Figure 3), only two studies [32,35] were on the right side because of their concomitant findings of both increases and decreases in different brain regions resulting in a lack of overall change in brain volume in individuals with excessive smartphone use (Figure 2). One possible explanation may be a difference in gender distribution. While there was a male predominance in the five studies on the left side of the funnel plot, one study on the left side did not report its gender ratio [35] and the female prevalence in the other [32] was up to 69%. Fourth, variations in the criteria for defining excessive smartphone use across the eligible studies may predispose to discrepancies in severity-related structural and functional alterations in their findings which, in turn, may bias our study outcomes. Finally, a previous study has shown an association of depression, anxiety, attention deficit hyperactivity disorder (ADHD), and excessive internet use with brain volume [8]. For instance, a decreased volume has been reported to be related to excessive internet use, which could be associated with depression, anxiety, and ADHD. Taking into account that excessive smartphone use is a complex process that may involve interaction of different emotional components, merely attributing the observed structural alteration to smartphone use would not be inappropriate. Moreover, whether our finding suggests a decreased brain volume requires further well-designed longitudinal cohort studies for elucidation.

## 5. Conclusions

Through systematically reviewing the currently available clinical trials focusing on structural and functional neural correlates in individuals with excessive smartphone use, the current meta-analysis not only demonstrated a reduced brain volume (especially subcortical regions) but also identified the brain areas with enhanced activities, namely the right posterior lobe of cerebellum, the right lingual gyrus (BA18), and the left middle frontal gyrus (BA46) in individuals with excessive smartphone use. Moreover, the structural impact of excessive smartphone use on brain volume was more pronounced in adolescents than in adults. In addition, there was a positive correlation between size reduction and impulsivity among individuals with excessive smartphone use. Due to the limited number of available clinical trials in this study, further investigations are warranted to verify our findings.

## Figures and Tables

**Figure 1 ijerph-19-16277-f001:**
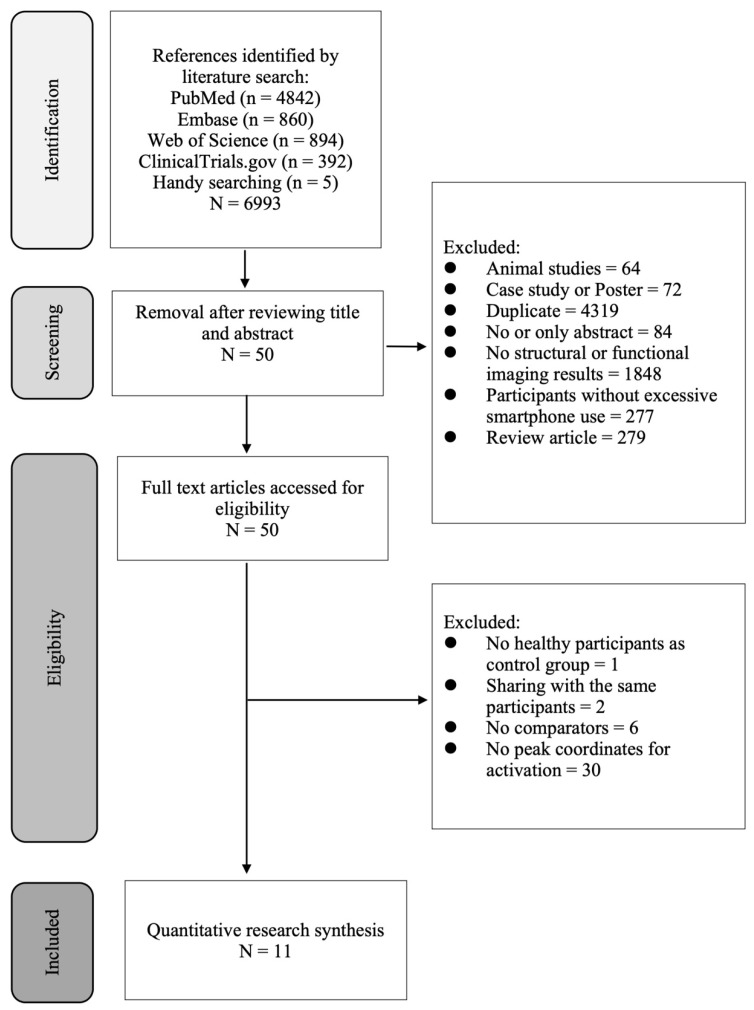
Flowchart summarizing the process of study selection.

**Figure 2 ijerph-19-16277-f002:**
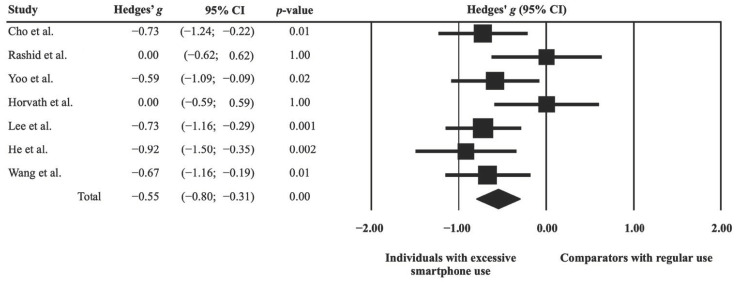
Forest plot comparing the effect sizes for brain volumes between individuals with excessive smartphone use and their comparators with regular use [32,33,35,36,38,41,42].

**Figure 3 ijerph-19-16277-f003:**
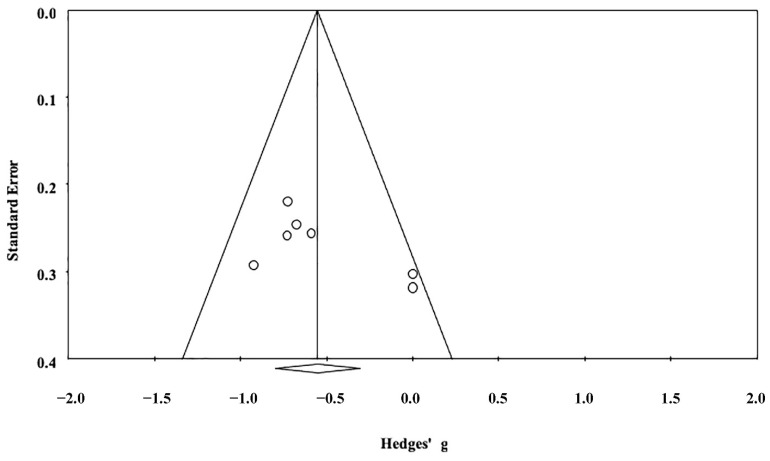
Random-effects funnel plot for detecting publication bias in studies reporting the brain volumes between individuals with excessive smartphone use and their comparators with regular use.

**Figure 4 ijerph-19-16277-f004:**
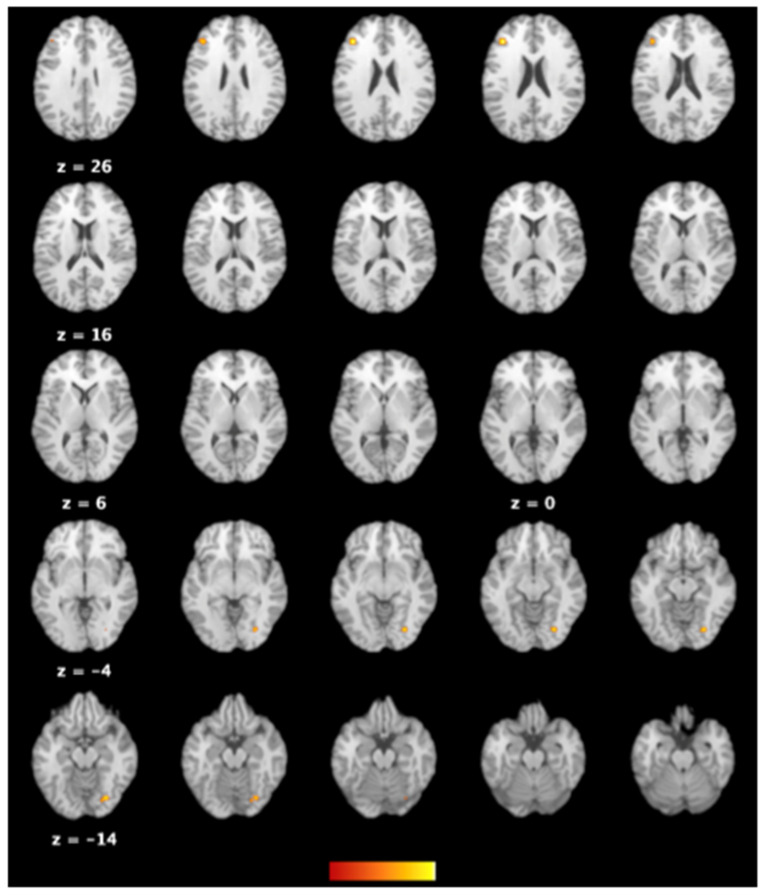
Functional magnetic resonance imaging (fMRI) studies showing brain regions with enhanced activities in individuals with excessive smartphone use compared to their comparators with regular use.

**Table 1 ijerph-19-16277-t001:** Comparison of structural and functional neural correlates between individuals with excessive smartphone use and their counterparts with regular use.

Studies	EXP			CON			EVAL of EXP	MPA Definition	Brain Volume			Functional Connectivity			Country
	N	% F	Avg. Age (SD)	N	% F	Avg. Age (SD)			ANAT Meas.	EXP < CON	EXP > CON	CM	EXP < CON	EXP > CON	
Cho et al. [33]	20	60.00	16.20 (1.11)	67	28.36	15.27(1.69)	SAPS	≥42 on SAPS as EXP	GMV and WMV	SCP	N/A	N/A			Korea
Choi et al. [34]	33	54.55	25.21 (5.54)	33	54.55	24.85(4.49)	SAPS	EXP score > CON score	N/A			BOLD signal	Left MOOC	Right: AG, FFG, SMG, MFG Left: SMA, supPL, PreCG, IFG, MCG, Precuneus Bilateral: ParaCG, IPL	Korea
Rashid et al. [35]	20	N/A	18–25	20	N/A	18–25	SAS-M	Subjects with scores of >98 were considered as EXP	GMD and GMV	Left IPL	Right insula and right PreCG	BOLD signal	N/A	Right: Calcarine gyrus, SFG, supMA, precuneus Left: FFG, supPL, PCC Bilateral: Cerebellum	Malaysia
Yoo et al. [36]	20	60.00	16.20 (1.11)	68	27.94	15.26 (1.68)	SAPS	A SAPS score of 42 and higher as EXP	GMV and WMV	Bilateral CN	N/A	N/A			Korea
Horvath et al. [32]	22	68.18	22.5 (3.0)	26	69.23	23.0 (3.2)	SAS-SV	scoring > 31 (M) and >33 (F) on SAS-SV	GMV	Left anterior insula, IT, and PHC	Left SMG	BOLD signal	Right ACC	N/A	Germany
Nasser et al. [37]	15	33.00	22.2 (0.86)	15	53.00	21.67 (1.18)	SAS-M and IGAT	A cut-off score of ≥98 as EXP	N/A			BOLD signal	Right: MOOC, infOG Left: Cuneus, supOG Bilateral: Calcarine, FFG, LG	Right: LG, supPL, IFG, IT, MOOC, MFG, postCG, SMG, SFG, MCG Left: FFG, MFG, MOOC, supPL, IPL, infOG, precuneus, insula, supOG, putamen, ACC, postCG	Malaysia
Lee et al. [38]	39	25.64	22.9 (2.2)	49	34.69	22.4 (2.7)	SAPS	Total SAPS score > 40, or subscale score > 14 for disturbance of adaptive function as EXP	GMV	Right OPFC	N/A	N/A			Korea
Lou et al. [39]	24	54.17	23.25 (1.33)	16	75.0	23.88 (0.86)	YIAT	Over 5 “yes” to the 8 questions as EXP	N/A			BOLD signal	N/A	Right: MTG, IT Bilateral: ACC, MFG, IFG	China
Chun et al. [40]	25	48.00	27.76 (5.97)	27	33.30	28.93 (6.93)	SAPS	Total > 44, or subscale > 15 (disturbance of adaptive function), >13(withdrawal), >13 (tolerance)	N/A			BOLD signal	Right: IT, supPL, dorsal ACC, SMA, cuneus, Thal Left: MFG, LG, MOOC, DLPFC, cerebellum, supOG Bilateral: preCG, MTG	N/A	Korea
He et al. [41]	25	32.0	24.12 (6.15)	25	32.0	29.80 (10.9)	FACIUI	EXP > CON	GMV	Right VS, bilateral amygdala	N/A	N/A			China
Wang et al. [42]	34	61.76	21.60 (2.10)	34	61.76	21.73 (1.94)	MPAI	Over 51 as EXP	GMV& DTI	Right: SFG, IFG, MOOC Left: ACC Bilateral: MFG, Thal, CgH(DTI)	N/A	N/A			China

ACC = anterior cingulate cortex; AG = Angular gyrus; ANAT = Anatomy; BOLD = Blood-oxygen-level-dependent; CM = Connectivity measure; CN = Caudate nucleus; CON = Control group; DLPFC = Dorsolateral prefrontal cortex; EVAL = Evaluation; EXP = Experimental group; F = Female; FACIUI = Facebook-specific adaptation of the Compulsive Internet Use instrument; FFG = Fusiform gyrus; GMD = Gray matter density; GMV = Gray matter volume; IFG = Inferior frontal gyrus; IGAT = Instagram addiction test; IPL = inferior parietal lobule; IT = Inferior temporal cortex; LG = lingual gyrus; M = Male; MA = mobile phone addiction; MCG = Middle cingulate gyrus; MFG = Middle frontal gyrus; MOOC = Middle occipital cortex; MPAI = Mobile phone addiction index; MTG = Middle temporal gyrus; N/A = not available; InfOG = Inferior occipital gyrus; OPFC = Orbitofrontal cortex; PCC = Posterior cingulate cortex; PHC = Parahippocampal cortex; PreCG = Precentral gyrus; postCG = Postcentral gyrus; SAPS = Smartphone addiction proneness scale; SAS-M = Smartphone addiction scale-Malay version questionnaire; SAS-SV = Short version of smartphone addiction scale; SCP = Superior cerebellar peduncle; SFG = Superior frontal gyrus; SMA = Supplementary motor area; SMG = Supramarginal gyrus; SPAI = Smartphone addiction inventory; supMA = Superior motor area; supOG = superior occipital gyrus; supPL = Superior parietal lobule; Thal = Thalamus; VS = Ventral striatum; WMV = White matter volume; YIAT = Young’s internet addiction test.

**Table 2 ijerph-19-16277-t002:** Comparison of brain volumes between individuals with excessive smartphone use and their counterparts with regular use: Hedges’ *g*.

Subgroup Analysis	N_Comp_	*g* ^a^	95% CI	Z	*Q* ^b^	*p* ^c^
Age group						
Adolescents	2	−0.66	−1.01 to −0.30	−3.61 **	0.41	0.00
Adults	5	−0.49	−0.85 to −0.14	−2.74 **		
Location						
Cerebral cortex ^d^	4	−0.14	−0.92 to 0.64	−0.34	2.35	0.00
Subcortical structure ^e^	5	−0.77	−1.01 to −0.54	−6.42 **		

_Comp_ Comparison; ^a^ According to the random effects model; ^b^ Cochran’s *Q* to measure the heterogeneity in accordance with random effects analysis; ^c^ The *P* levels in this column indicate whether the difference between the effect sizes in the subgroups is significant; ^d^ Encompassing Rashid et al. [35], Horvath et al. [32], Lee et al. [38], Wang et al. [42]; ^e^ Consisting of Cho et al. [33], Yoo et al. [36], Horvath et al. [32], He et al. [41], Wang et al. [42]. ** *p* < 0.01.

**Table 3 ijerph-19-16277-t003:** Regression analysis of correlations between different mediators and changes in brain volume among individuals with excessive smartphone use.

Variable (Continuous)	Coefficient (95% CI)	*p*
Prevalence of female	0.01 (−0.002 to 0.03) (*n* = 6)	0.09
Age	−0.006 (−0.07 to 0.06) (*n* = 6)	0.86
Intelligence quotient	−0.006 (−0.07 to 0.06) (*n* = 3)	0.85
Mean Beck depression inventory score	−0.08 (−0.36 to 0.19) (*n* = 3)	0.55
Mean Barratt impulsiveness scale score	0.03 (0.004 to 0.06) (*n* = 4)	0.03

**Table 4 ijerph-19-16277-t004:** Brain areas with increased activities in participants with excessive smartphone use compared with those with regular use.

Cluster	Side	Brain Area	BA	Volume (mm^3^)	ALE	x	y	z
1	R	Declive of posterior lobe		568	0.014351934	28	−76	−14
1	R	Lingual gyrus	18		0.014068902	28	−76	−8
2	L	Middle frontal gyrus	46	472	0.016162576	−38	30	20

Note: All areas reported in Talairach space; Broadmann areas are defined by Brain Map Talairach atlas; ALE = activation likelihood estimation; BA = Broadmann area; L = Left; R = Right.

## Data Availability

Not applicable.

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
