# Peer review of "Structural and Functional Neural Correlates in Individuals with Excessive Smartphone Use: A Systematic Review and Meta-Analysis"

_ijerph, 2022, doi:10.3390/ijerph192316277_

Round 1
Author Response
Responses to Reviewer 1’s Comments
Summary
The submitted manuscript focused on a comparison of brain volumes and activations between excessive smartphone users and individuals with regular use based on magnetic resonance imaging using a meta-analytical approach. The results demonstrated a smaller brain volume in excessive smartphone users than that in the controls especially in subcortical regions and in adolescents. In addition to their finding of a significant positive association between impulsivity and volume reduction, the authors showed altered activations in the declive of the posterior lobe of cerebellum, the lingual gyrus, and the middle frontal gyrus in excessive smartphone users. As a whole, the manuscript presents quite interesting and novel findings. It was also well-written with the inclusion of important references in this field. However, there are some important points that the authors need to address before being considered for publication. Besides, some minor points about language usage need to be paid attention to.
Response:
We would like to thank the Reviewer for all the professional suggestions and inquiries, in compliance with which we have made point-by-point responses as follows. Please kindly note that the corresponding changes in the revised manuscript are marked in red.
General concept comments:
Comment 1:
For a meta-analysis, the number of included studies is adequate (i.e., more than 10) with the inclusion of close to seven thousands subjects. The collection of literature and the analyses were complete. Overall, the most notable strength of the study is that, on top of a volumetric investigation of the brain regions affected, the authors provided evidence-based information on the areas of activation based on functional MRI studies and offered possible explanations from previous studies to address the knowledge gap in this specific field. I think the authors did a good job.
Response 1:
We are grateful to the Reviewer for the highly encouraging comments.
Comment 2:
Nevertheless, the authors claimed that the study followed the PRISMA guidelines but I could not find a PRISMA checklist, which I think should accompany the submission maybe as a supplement.
Response 2:
Accordingly, we have added the PRISMA guidelines as Supplemental Table 1 of the revised manuscript.
Specific Comments:
Major Comments:
Comment 3:
As the authors pointed out that excessive smartphone use has not been classified
into a distinct disorder, the authors should provide more references about the definition of “excessive smartphone user” as well as the rating scales that the studies used in order to justify the use of this term in their study. Because the authors already mentioned this as a limitation that there was a variation in this definition in their included studies, they should provide more information about the differences in definition in their included studies.
Response 3:
In response to the Reviewer’s insightful comment, we have summarized the definitions of “excessive smartphone user” in Supplemental Table 2 of the revised manuscript.
Comment 4:
Despite the same amount of time devoted to smartphone use, the excessive users may utilize their devices for different purposes such as communication, game playing, and even for browsing pornographic materials that would affect different regions of the brain both anatomically and functionally. Therefore, the term “excessive smartphone use” may be too generalized. I’m just wondering if relevant information is available in the authors’ included studies. If not, the authors should at least include the information in the limitation section to make it clear to the readers.
Response 4:
We completely agree with the Reviewer that the purposes of smartphone use are likely to trigger various emotional responses and hence the activation/suppression of different brain regions. Despite the consistency of our findings, we have reinforced this information into the limitation section of the revised manuscript.
Comment 5:
The terminology of this paper is not clear. For example, the introduction refers to smartphone addiction, later on to excessive smartphone use. The term smartphone addiction was and still is heavily debated. Therefore, the terminology in this manuscript needs to be consistent.
Response 5:
In compliance with the Reviewer’s professional comment, the term “smartphone addiction” has been replaced by “excessive smartphone use” throughout the manuscript.
Comment 6:
The authors should provide their search strategy to allow reproduction of their results for the readers.
Response 6:
Accordingly, we have included our search strategy into Supplemental Table 3 of the revised manuscript.
Comment 7:
The author should explain the use of random effect models for minimizing sample size bias.
Response 7:
We are thankful to the Reviewer’s comment. We have highlighted the detail about the use of random-effects models in the Methods section (subsection 2.4. “Effect size analysis”) of the revised manuscript.
Comment 8:
Regarding the references cited, all are relevant to the topic of the manuscript. There is no self-citation. However, out of the 62 references, only 24 (38.7%) were published in the last five years. I would suggest the inclusion of at least 31 recent references (i.e., within the last five years) to ensure the novelty of information in this study.
Response 8:
Accordingly, we have reviewed and replaced the older references so that there are now 31 references (out of a total of 61) published within the last five years.
Minor Comments:
Comment 9:
Abstract: Because this meta-analysis also included “Original, cross-sectional comparative studies” as stated in the inclusion criteria (section 2.1), the use of the term “clinical trial” is incorrect.
Response 9:
We would like to thank the Reviewer for the meticulous observation. Accordingly, the term “clinical trials” has been replaced by “observational studies”.
Comment 10:
Abstract: Probability values should be presented in italics. The same should be applied to the whole manuscript.
Response 10:
Accordingly, all probability values have been presented in italics.
Comment 11:
Introduction (2nd paragraph): In the sentence “In contrast to T2-weighted imaging, the T1-weighted approach which provides the greatest clarity in distinguishing among gray matter...”, the use of “which” is inappropriate taking into account the difference between restrictive and nonrestrictive clauses. It should be replaced by “that”.
Response 11:
We are grateful to the Reviewer for the insightful reminder. We have changed “which” to “that”.

Reviewer 2 Report
The study entitled “Structural and Functional Neural Correlates in Individuals With Excessive Smartphone Use: A Systematic Review and Meta-Analysis” used a systematic review and meta-analysis examine the differences in brain volumes and activations between excessive smartphone users and individuals with regular use by magnetic resonance imaging. The topic is timely and important because prior evidence shows that there are more and more people who have the problem of excessive use of smartphones. The findings demonstrated a potential association of excessive smartphone use with a reduced brain volume and altered activations.
However, I still have a few suggestions for authors.
Introduction
1. The introduction section should provide information on whether there has been studies similar to this study in the past, and state the importance of this study. The current content fails to see why it is important to conduct this research.
Methods
1.Since the keywords of this study include "non" smartphones in the search strategy (such as mobile phone), however, this study focuses on the impact of smartphones. Did the authors exclude article using mobile phones (non-smart)?
Results
1. Based on the above question, there are two articles using CIUS and Young's internet addiction test respectively. And are their participants using smartphones only?
Discussion
Discussion (including limitation) is well oriented.
Overall, this paper is well-written, and it has an original objective and a meaningful content. I congratulate the authors for the work done.
Author Response
Responses to Reviewer 2’s Comments
General Comments:
The study entitled “Structural and Functional Neural Correlates in Individuals With Excessive Smartphone Use: A Systematic Review and Meta-Analysis” used a systematic review and meta-analysis examine the differences in brain volumes and activations between excessive smartphone users and individuals with regular use by magnetic resonance imaging. The topic is timely and important because prior evidence shows that there are more and more people who have the problem of excessive use of smartphones. The findings demonstrated a potential association of excessive smartphone use with a reduced brain volume and altered activations. However, I still have a few suggestions for authors.
Response:
We are sincerely grateful to the Reviewer for the valuable time taken to review our manuscript, the encouraging comments as well as all the professional suggestions and inquiries, in compliance with which we have made point-by-point responses below. Please kindly note that the corresponding changes in the revised manuscript are marked in blue.
Comment 1:
Introduction: The introduction section should provide information on whether there has been studies similar to this study in the past, and state the importance of this study. The current content fails to see why it is important to conduct this research.
Response 1:
In response to the Reviewer’s professional inquiries, we have reinforced the description in the Introduction section of the revised manuscript. Indeed, our study is the first to address the associations of excessive smartphone use with anatomy of brain structures and brain activities. In compliance with the Reviewer’s important comment, we have highlighted this information in Introduction with the inclusion of an additional reference.
Please cite as a reference: https://www.bankmycell.com/blog/how-many-phones-are-in-the-world (access date: November 26, 2022)
Comment 2:
Methods: Since the keywords of this study include "non" smartphones in the search strategy (such as mobile phone), however, this study focuses on the impact of smartphones. Did the authors exclude article using mobile phones (non-smart)?
Response 2:
We are thankful to the Reviewer’s insightful comment. We did exclude articles investigating “non-smart” mobile phones through manual screening. In response to the Reviewer’s key inquiry, we have added this as one of our exclusion criteria in the current study (Methods, 2.1. Study eligibility). In addition, we have incorporated our literature search strategy (Supplemental Table 2) into the revised manuscript to clarify the information to the readers and to ensure reproducibility of our search results.
Comment 3:
Results: Based on the above question, there are two articles using CIUS and Young's internet addiction test respectively. And are their participants using smartphones only?
Response 3:
The Reviewer’s insightful comment is highly appreciated. The first article used CIUS to assess smartphone users’ Facebook addiction. The second article first adopted the internet addiction assessment tool, Young’s addiction test, to divide their study participants into internet addiction and non-internet addiction groups, followed by comparing their degrees of smartphone addiction with the smartphone addiction proneness scale. Both studies focused on smartphone users and were considered eligible for the current study.
Comment 4:
Discussion: Discussion (including limitation) is well oriented.
Response 4:
We would like to thank the Reviewer for the encouraging comment.
Comment 5:
Overall, this paper is well-written, and it has an original objective and a meaningful content. I congratulate the authors for the work done.
Response 5:
We are truly thankful to the Reviewer for the highly encouraging comment and the professional suggestions that substantially improved the quality of our work.
